# Two-Year Study on the Intra-Individual Dynamics of Gut Microbiota and Short-Chain Fatty Acids Profiles in Healthy Adults

**DOI:** 10.3390/microorganisms12081712

**Published:** 2024-08-20

**Authors:** Anastasia Senina, Maria Markelova, Dilyara Khusnutdinova, Maria Siniagina, Olga Kupriyanova, Gulnaz Synbulatova, Airat Kayumov, Eugenia Boulygina, Tatiana Grigoryeva

**Affiliations:** 1Institute of Fundamental Medicine and Biology, Kazan Federal University, 420008 Kazan, Russia; anastasiahm@list.ru (A.S.); mimarkelova@gmail.com (M.M.); dilyahusn@gmail.com (D.K.); marias25@mail.ru (M.S.); olgakupr2010@mail.rugulnazgg12@gmail.com (G.S.); kairatr@yandex.ru (A.K.); boulygina@gmail.com (E.B.); 2Regional Research and Testing Center “Pharmexpert”, Kazan State Medical University, 420012 Kazan, Russia

**Keywords:** gut microbiota, 16S rRNA, short chain fatty acids (SCFAs), healthy microbiota, temporal stability, seasonal changes, individual fingerprint

## Abstract

While the gut microbiome has been intensively investigated for more than twenty years already, its role in various disorders remains to be unraveled. At the same time, questions about what changes in the gut microbiota can be considered as normal or pathological and whether communities are able to recover after exposure to negative factors (diseases, medications, environmental factors) are still unclear. Here, we describe changes in the gut microbiota composition and the content of short-chain fatty acids in adult healthy volunteers (*n* = 15) over a 24 month-period. Intraindividual variability in gut microbial composition was 40%, whereas the short chain fatty acids profile remained relatively stable (2-year variability 20%, inter-individual 26%). The changes tend to accumulate over time. Nevertheless, both short-term and long-term changes in the gut microbiome composition were significantly smaller within individuals than interindividual differences (two-year interindividual variability was 75%). Seasonal changes in gut microbiota were found more often in autumn and spring involving the content of minor representatives (less than 1.5% of the community in average) in the phyla *Actinobacteriota*, *Firmicutes* and *Proteobacteria*.

## 1. Introduction

The human gut is an excellent habitat for a wide variety of microorganisms that evolve and adapt to their host throughout the host’s life [1]. The human gut microbiota is involved in several processes. It regulates intestinal and systemic hormones [2], proliferation and differentiation of epithelial cells, modification and elimination of specific toxins/drugs [3,4], bone growth and development [5,6], modulation of intestinal barrier function [7,8], and sleep duration and quality [9]. It has been shown that dysbiosis in the gut microbiota is associated with various disorders, including inflammatory bowel disease, obesity, non-alcoholic fatty liver disease, atherosclerosis [10], type 2 diabetes [11], cancer [11], cardiovascular diseases, and nervous system diseases [11] such as depressive disorder, schizophrenia, bipolar disorder, autism spectrum disorder, and attention deficit hyperactivity disorder [11,12,13,14,15,16,17,18]. However, the exact role of microbiota dysbiosis in these pathologies is not well understood [11].

As one of the most important activities, gut microorganisms metabolize complex nutrients such as plant cell wall components (cellulose, pectin, hemicellulose, lignin) to simple sugars and then to short-chain fatty acids (SCFAs). These metabolites affect host health at the cellular, tissue, and organ levels via anti-inflammatory effects on the intestinal mucosa, histone acetylation, as well as gene regulation of cell proliferation, differentiation, and inflammatory response [19,20]. Thus, members of the phylum *Bacteroidetes* produce mainly acetate and propionate, while the families *Ruminococcaceae* and *Lachnospiraceae* of the phylum *Firmicutes* produce butyrate. Genomic profiling of the gut bacteria indicates that members of the phyla Actinobacteria, Bacteroidetes, Fusobacteria, Proteobacteria, Spirochaetes and *Thermotogae* are also potential butyrate producers [21]. The imbalance in the gut microbiota results in decrease of SCFAs levels leading to chronic inflammation and metabolic dysfunction [22], cancer [11], cardiovascular diseases [13], and nervous system diseases [12,22]. Most studies have focused on the relationship between gut microbiota, its metabolites, and human health [11,23,24,25], and there are only a few reports on the intraindividual variability of fecal SCFAs [21].

It has been shown that some bacteria of a core community persist in healthy adults for many months and even years, despite regular fluctuations in their abundance [26,27,28,29,30,31,32]. In addition, high microbial diversity has been shown to be associated with gut community stability [33]. Conversely, some studies have reported wide variability of gut microbiota in healthy adults over time [34,35,36]. According to a recent study, as much as 23% of total microbiota variation is intraindividual [37]. Notably, colonization efficacy depends on individual characteristics of both the host and microorganisms, which have different competitive abilities [38,39,40,41]. Therefore, a better understanding of normal microbiota may hold a key to its correction through dietary interventions, probiotics, prebiotics, and antibiotics [25,42,43,44]. Thus, the gut microbiota is a complex and dynamic ecosystem, but understanding its plasticity remains challenging. In this study, we investigate long-term changes in gut microbiota composition and SCFA production in healthy volunteers in the Volga region of Russia over two years, including assessment of the seasonal effects.

## 2. Materials and Methods

### 2.1. Design of the Study

The study included 15 healthy volunteers (men and women aged 26–52 years) (Appendix A) for investigation of intra-individual variability in fecal microbiome (biodiversity on base 16S rRNA gene) and SCFAs profiles. These healthy volunteers were recruited from Kazan (Volga region of Russia) who did not meet the following exclusion criteria:Refusal to give informed consent.Presence of diseases/conditions such as recent surgery, current cancer or infectious disease, inflammatory bowel disease, acute relapse of any chronic disease, mental illness, type 1 and 2 diabetes, malabsorption syndrome associated with established disease of the small intestine or pancreas.Immunodeficiency conditions.Presence of diarrhea (with a stool frequency of more than 3 times a day) for at least 3 consecutive days during the last month.Taking of probiotics, prebiotics, antibiotics, cytostatics, glucocorticosteroids, and immunosuppressants 6 months before the start of the study.Donation of blood or its components within three months before the start of the study.Regular smoking and/or alcohol consumption (2 weeks prior to screening).Pregnancy or lactation.

Informed consent was obtained from all subjects involved in the study. The study was conducted in accordance with the recommendations of the local ethics committee of the Kazan Federal University, Kazan, Russia (Protocol No. 6, dated 13 October 2017). All individuals included in the study maintained their general lifestyle, including dietary and physical activity habits, throughout the study period.

In total, at least 10 samples were collected from each volunteer over two years starting in February 2021 (Figure 1).

Due to antibiotics admissions 3 individuals were excluded during the course of the study.

### 2.2. 16S rRNA Sequencing and Analysis

Stool DNA was extracted using the MP FastDNA Spin Kit for Feces (MP Biomedicals, CA, USA). DNA concentration was determined using a Qubit 2.0 fluorometer (Life Technologies, Carlsbad, CA, USA). Amplification of the 16S rRNA gene V3–V4 (forward primer 341f: 5′-TCGTCGGCAGCGTCAGATGTGTATAAGAGACAGTCGTCGGCAGCGTCAGATGTGTATAAGAGACAGCCTACGGGNGGCWGCAG-3′; reverse primer 806r: 5′-GTCTCGTGGGCTCGGAGATGTGTATAAGAGACAGGTCTCGTGGGCTGGAGATGTGTATAAGAGACAGGACTACHVGGGTATCTAATCC-3′) was performed according to a standard protocol (Illumina, San Diego, CA, USA), amplicons were purified using AMPure XP beads (Beckman Coulter, Brea, CA, USA) and indexed and repurified according to the manufacturer’s protocol. The concentrations of PCR products were quantified using the Qubit dsDNA HS Assay Kit (Thermo Scientific, Waltham, MA, USA) and pooled in equimolar amounts. Quality of the libraries was analyzed with a Bioanalyzer DNA 1000 Chip. Further analysis of the nucleotide sequence of 16S rRNA amplicons was carried out using the MiSeq platform (Illumina, San Diego, CA, USA) with v3 reagents (2 × 300 cycles). Raw reads were processed using the DADA2 algorithm implemented in QIIME [45]. After quality filtering and removal of chimera and phiX sequences, we analyzed N joined read pairs per sample on average. The taxonomy was assigned to the sequences based on OTUs with a 99% similarity threshold using a Naïve Bayes classifier pre-trained on the Greengenes2 (v.2022.10) database [46] P. To characterize the richness and evenness of the bacterial community, alpha diversity indices were calculated using Shannon’s and Faith PD metrics. Raw reads were deposited in the SRA under Project ID PRJNA1067295 in the fastq format (https://www.ncbi.nlm.nih.gov/bioproject/1067295, accessed on 21 January 2024).

### 2.3. Determination of SCFA in Feces

Sample preparation was carried out according to Niccolai et al. [47]. Fecal samples were collected in falcon tubes and stored at −80 °C. To 0.5–1.0 g of sample, 10 mM sodium bicarbonate solution (1:1 *w*/*v*) was added in a 1.5-mL centrifuge tube, mixed in a vortex apparatus, extracted in an ultrasonic bath (5 min), and then centrifuged at 5000 rpm (10 min). The supernatant was collected in a separate 1.5 mL tube. Finally, SCFAs were extracted as follows: 1 mL tert-butyl methyl ether and 50 μL 1.0 M HCl solution were added to a 100 μL aliquot of sample solution. After this, each time the tube was vortexed for 2 min, centrifuged at 10,000 rpm for 5 min, and finally the solvent layer was transferred to an autosampler vial and analyzed using a Clarus 680 gas chromatograph (Perkin Elmer, Waltham, MA, USA) with a flame ionization detector [48]. An HP-FFAP capillary column (50 m × 0.32 mm × 0.50 µm) with poly (ethylene glycol) modified with nitroterephthalic acid (CAS number 19091F-115, Agilent Technologies, Santa Clara, CA, USA) was used for the chromatographic separation. A glass liner stoppered with a glass wool plug in the middle of the liner was used to prevent contamination of the GC column with nonvolatile fecal materials. The temperature of the column was set initially to 100 °C for 1 min and increased to 200 °C at a rate of 10 °C/min, which was maintained for 6 min. The total run time was 17 min. The injection temperature was 230 °C, and the detector temperature was 220 °C. Nitrogen was used as a carrier gas in a constant flow mode; the flow rate was 1.3 mL/min. The make-up gas (N2) was at 20 mL/min. The flow rate of H2 and air were 45 mL/min and 450 mL/min, respectively. The injection was carried out using an autosampler with a Perkin Elmer 5 µL microsyringe (Perkin Elmer, Waltham, MA, USA). For analysis, 2 μL of the sample solution was injected in split mode at a ratio of 20:1. The amount of SCFAs in the sample solution was determined using TotalChrom ver. 6.3.2 software (Perkin Elmer, Waltham, MA, USA) by the external-standard method.

### 2.4. Statistical Analysis

Differences in Bray-Curtis dissimilarity between different samples were assessed using the Kruskal-Wallis test. All statistical calculations were performed using the R v.4.2.2 programming language in the RStudio v.2023.12.1 + 402 program. A *p* value of <0.05 was considered significant.

## 3. Results

### 3.1. Gut Microbial Community Analysis

Dynamic changes in gut microbial taxonomic composition were assessed based on 16S rRNA gene sequencing. The study design allows us to assess both short-term (2 weeks–1 month) and long-term intraindividual changes in community structure, as well as interindividual variability.

#### 3.1.1. Alpha Diversity Indices

Initial Shannon index values varied from 5.6 to 6.8 (mean ± SD 6 ± 0.4), and observed features from 165 to 360 (mean ± SD 256 ± 65) (Figure 2).

In our study, Shannon index over the entire two-year study period varied from 4.98 to 7.2 (mean ± SD 6.2 ± 0.4), and observed features varied from 144 to 407 (mean ± SD 251.7 ± 57.6) (Appendix A).

Of the 10 samples collected over two years, four were collected during the winter period, and two each were collected during the remaining seasons. Common trends in alpha diversity indices associated with seasonal changes were not detected (Figure 3).

#### 3.1.2. Beta Diversity Indices

Beta diversity reflects the similarity between different samples [49,50]. Samples from each participant were grouped into clusters based on the Bray-Curtis distance (Figure 4A–C). Samples from the same participant had significantly lower Bray-Curtis dissimilarity values than samples from different participants (0.45 and 0.75, respectively, *p*-value < 0.05) (Figure 4D). Thus, at different points in time, intra-individual microbiota variability was less than inter-individual variability, suggesting the relative stability of an individual’s unique microbiota profile.

Significant differences were observed between intraindividual Bray-Curtis distance values for samples taken two weeks, one year, and two years apart (2 w—36%, 1 y—46%, 2 y—58%) (Figure 5). So, the changes tend to accumulate over time.

#### 3.1.3. Taxonomic Analysis

In this study the dynamics of the relative abundances of bacterial phyla and families were analyzed (Figure 6). The most abundant phyla were *Firmicutes_A* (58.5 ± 11.3%), *Bacteroidota* (22.5 ± 10.9%), *Actinobacteriota* (6.4 ± 5.9%) and *Firmicutes_D* (5.6 ± 6.7%). Among bacterial families, the most dominant were *Lachnospiraceae* (28.0 ± 9.6%), *Bacteroidaceae* (16.2 ± 10.7%) and *Ruminococcaceae* (14.0 ± 5.7%). The microbiome of one volunteer (Figure 6B, participant 3) was dominated by the *Streptococcaceae* family over 10 months (mean 29.5% ± 0.06%); the proportion of *Streptococcacea* subsequently decreased significantly to near the average among other participants (0.99 ± 1.2%). Some participants had an increased proportion of the family *Bifidobacteriaceae*. The microbiota of one study participant (Figure 6A,B, participant 1) was characterized by increased presence of phylum *Actinobacteriota* (20.7 ± 6.6% to 5.4 ± 4.4%), represented mostly by the species *Bifidobacterium adolescentis* (11.5 ± 0.4%). These trends correspond to periods of daily consumption/exclusion of fermented milk products according to participants’ diet diaries.

### 3.2. Short-Chain Fatty Acids

In this study, we quantified the major SCFAs described in the human intestine. The most abundant were acetate (1.34 ± 0.6 mg/g), which varied from 0.3 to 3.3 mg/g, propionate (0.45 ± 0.2 mg/g), which varied from 0.1 to 1.1 mg/g, and butyrate (0.4 ± 0.3 mg/g), which varied from 0.05 to 1.5 mg/g. The amounts of all SCFAs measured for each participant are presented in Appendix A. The variability of the measured SCFAs (minimum and maximum values) is shown in Table 1. Intra-individual Bray-Curtis values were lower than inter-individual values for all SCFA levels studied (Figure 7), indicating that intra-individual variability was less than inter-individual variability.

### 3.3. Seasonal Changes in the Bacterial Community and SCAFs of the Gut Microbiota

We found a few statistically significant seasonal changes in the abundance of several bacterial genera and families (Table 2) that comprise less than 1.5% of the gut community (on average). The relative abundance of the *Pseudomonadaceae* family increased in spring samples compared to other seasons; the genus *Pseudomonas_F* was absent in winter and autumn samples, and its abundance in spring was higher than in summer samples. In spring, compared to other seasons, the proportion of the genus *Klebsiella_724518* from the *Enterobacteriaceae* family also increased. The family *Veillonellaceae* increased in spring samples compared to winter samples. Bacteria of the family *Moraxellaceae* were absent in winter and autumn samples, and in summer samples they were higher than in spring samples. Compared to summer samples, the proportion of the family *Eggerthellaceae* was higher in spring samples but lower in autumn ones. And the abundance of the genus *Anaerobutyricum* of the family *Lachnospiraceae* decreased in autumn compared to other seasons.

The variability in Shannon’s alpha diversity index was not associated with seasonal changes. Significant changes were found only for the Faith PD index, the values of which were lower in spring compared to autumn and summer.

Seasonal changes in SCFA levels are presented in Appendix A. Significant differences were found only for valeric acid. Its level was increased in the summer samples (0.14 ± 0.09 mg/g) compared to the autumn samples (0.09 ± 0.04 mg/g).

## 4. Discussion

There is growing interest in the connection between the composition of the human gut microbiota and various pathologies [11,15,17,51,52]. To better understand the role of the gut microbiota in pathogenesis, it is necessary to determine how stable the bacterial community is over time, and how intra-individual microbial differences and exposure to adverse factors relate to health status. In our study, we characterized the gut microbiota of healthy volunteers (*n* = 15) over a 24-month period. The first set of samples was collected every two weeks, then every 2–3 months. Alpha diversity reflects the richness and evenness of the bacterial community. Summarizing and comparing community composition based on alpha diversity is a universal approach to microbial community analysis [49,50]. In our study, alpha diversity indices varied widely, and their variability was not associated with seasonal changes. There are only a few studies indicating that gut microbiota biodiversity indices are correlated with seasonality, but these changes have been directly linked to dietary factors, as has been shown in macaque monkeys [53] and closed human populations such as the Hutterites [54]. It has previously been shown that individuals with higher alpha diversity have less variation in the microbial composition of the gut community over time [55]. In our case, there was no relationship between the stability of microbial composition and initial diversity indices; it is likely that the spread of these indices, which was on the order of 20–25% between volunteers in our study, is not critical and sufficient to identify such a trend.

By comparing intra- and inter-individual Bray-Curtis values, we found that differences in the gut microbiota during the study were smaller within individuals than between participants. According to Bray-Curtis dissimilarity, each participant’s samples were clustered is PCoA plots, which is consistent with previous studies about intra-individual stability of gut microbiome [56,57]. According to a recent study of the Swedish population, up to 23% of the total variation in the microbiota was intra-individual over the 1-year study period [37]. In our study intra-individual variability of taxonomic composition was 46% and 58%, for 1-year and 2-year periods, respectively. These differences may be related to population, geographic, and methodological aspects [58,59].

Healthy human gut communities commonly contain two dominant bacterial phyla: *Firmicutes* and *Bacteroidota* [60,61,62]. Many researchers have used the *Firmicutes/Bacteroidota* ratio (*F/B* ratio) to characterize the gut microbiota and associate it with various host pathologies [61,63,64,65]. For example, the gut microbiota of obese individuals has a higher *F/B* ratio compared to that of normal-weight individuals [62,63,64]. Patients with non-gastrointestinal cancers, diarrhea after poisoning, recovering from antimicrobial therapy, with inflamed bowel disease, and smokers have a lower *F/B* ratio and Shannon diversity values compared to healthy individuals of the same age [65,66,67]. Also, a decrease in *Firmicutes* and an increase in *Bacteroidota* was detected in samples from patients with Alzheimer’s disease [18]. However, a growing number of studies question the use of *F/B* ratio as a marker for various pathologies [68,69,70]. The relative representation of *Firmicutes* and *Bacteroidota* varies greatly among individuals from the same population. This is due to the influence of various lifestyle factors on the composition of the gut microbiota. In our study, the ratio between *Firmicutes_A* (58.5 ± 11.3%) and *Bacteroidota* (22.5 ± 10.9%) varied not only among participants but even at different time points within the same individual. Maximal individual variability, measured as the difference between the maximum and minimum values, reached up to 44.4% for *Bacteroidota*, 39.8% for *Firmicutes_A*, 37.2% for *Firmicutes_D*, and 10.2% for *Firmicutes_C*.

We showed individual microbiota patterns can be related to the participant’s lifestyle. For instance, one individual was predominantly colonized by *Streptococcaceae* for 10 months. It has been shown that such an increase can be associated with obesigenic/high-fat diet and various inflammatory bowel pathologies [71,72]. In our study, the abundance of bacteria may have been due to the participant’s diet, which included streptococcal fermented foods. Species of the family *Streptococcaceae* are used as a starter culture for yogurt and many cheeses [73,74]. A recent study showed that *Streptococcaceae* increased in individuals who consumed 30 g of “Brynza” cheese per day [75].

The gut microbiome of another participant in the study who consumed fermented milk products daily, mainly kefir, was characterized by an increase in the abundance of the phylum *Actinobacteriota* (20.7 ± 6.6% to 5.4 ± 4.4%), represented mainly by the species *B. adolescentis* (11.5 ± 0.4%), which is known as a probiotic [76,77,78]. In April 2021, the proportion of the phylum *Actinobacteriota* in the microbial community of the study participant reached up 31%.

One of the key features of gut microbiota is the consumption fiber (complex carbohydrates) that is not digestible by humans and produces short chain fatty acids (SCFAs) as a major product. This process is important for the beneficial effects of prebiotics on human health. In recent years, SCFAs have been considered as a biomarker of human gut health and metabolism [79]. They improve barrier function, reduce intestinal inflammation, and affect transit and peristalsis [19,22,79,80].

Studies on modulating the gut microbiota in healthy individuals by increasing SCFA levels have shown mixed results [79]. Overdose of butyrate induces apoptosis and reduces the number of viable Caco-2 cells in a dose-dependent manner [81].

In order to develop a strategy for lifelong host health, it is important to determine fecal SCFA levels in healthy individuals. In this study, we quantified all of the most important fecal SCFAs whose concentrations varied widely: acetate varied from 0.3 to 3.3 mg/g, propionate—from 0.1 to 1.1 mg/g, and butyrate—from 0.05 to 1.5 mg/g. The normal level of SCFAs in the human gut is still uncertain. Data varies greatly depending on the SCFA extraction method used [82,83,84,85].

Seasonal variations in the gut microbiome composition have been studied less frequently than the influence of other factors. A few studies have looked at seasonal changes in the Hadza hunter-gatherers group, who change their diets between the rainy and dry seasons. This leads to a dramatic change in the composition of the gut microbiome [86].

However, the availability of staple foods throughout the year reduces the influence of season on diets in industrialized countries. For example, the composition of the gut microbiota of healthy Japanese did not fluctuate throughout the year, but seasonal changes were observed in patients with IBD and celiac disease, possibly related to seasonal exacerbations. Thus, the proportion of the phyla *Actinobacteria* and *TM7* was significantly higher in autumn compared to spring and winter [87].

Even the increased fiber intake in summer was not associated with an increase in certain bacterial taxa in the gut microbiota. It has been reported that the abundance of *Bacteroidota* was reduced in summer samples, while *Actinobacteria* increased [88,89]. In our study, a few statistically significant seasonal changes affected bacterial genera and families that made up less than 1.5% of the community on average. The changes affected not only families and genera of *Actinobacteriota*, but also *Firmicutes* and *Proteobacteria*, mainly in autumn and spring samples. Variability in alpha diversity was not associated with seasonal changes.

There are several limitations to this study. Because it was a single-center study with a small cohort size, and a preponderance of female volunteers, interindividual gut microbiota diversity and dietary factors could not be sufficiently characterized. The trends in intra-individual dynamics that were observed can be verified by increasing cohort size and expanding the geography of participants.

## 5. Conclusions

Our study showed that intra-individual patterns of gut microbiota were observed during the 24-months study period. Seasonal changes affected minor (non-dominant) bacterial genera and families. In addition, despite wide inter-individual taxonomic diversity in gut microbial composition and its dynamic changes, the levels of SCFAs measured in this study will help to determine the range associated with long-term health. Thus, the study contributes to the understanding of normal gut microbiota.

## Figures and Tables

**Figure 1 microorganisms-12-01712-f001:**
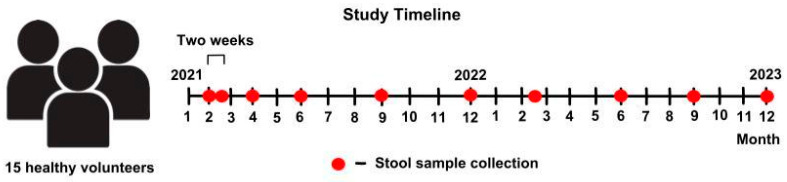
The study timeline.

**Figure 2 microorganisms-12-01712-f002:**
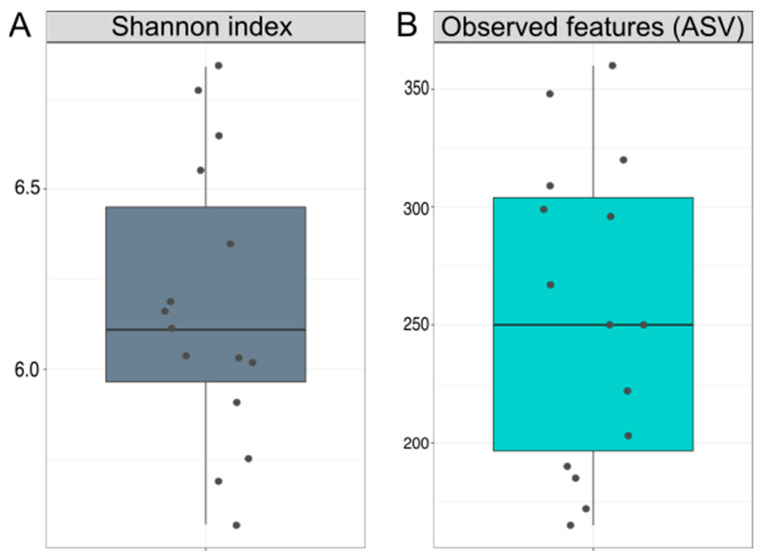
Box-plot of initial alpha diversity indices: (**A**) Shanon indices and (**B**) observed features (ASV).

**Figure 3 microorganisms-12-01712-f003:**
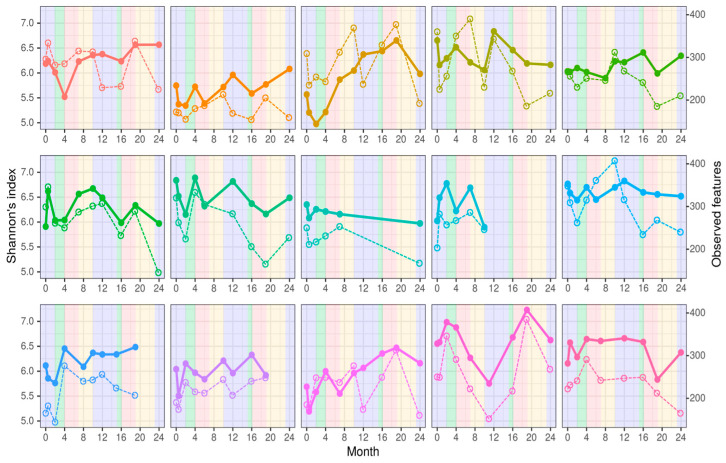
Dynamics of Shannon index values (continuous lines) and ASV counts (dashed lines). The background color reflects the season: blue—winter, green—spring, red—summer, yellow—autumn).

**Figure 4 microorganisms-12-01712-f004:**
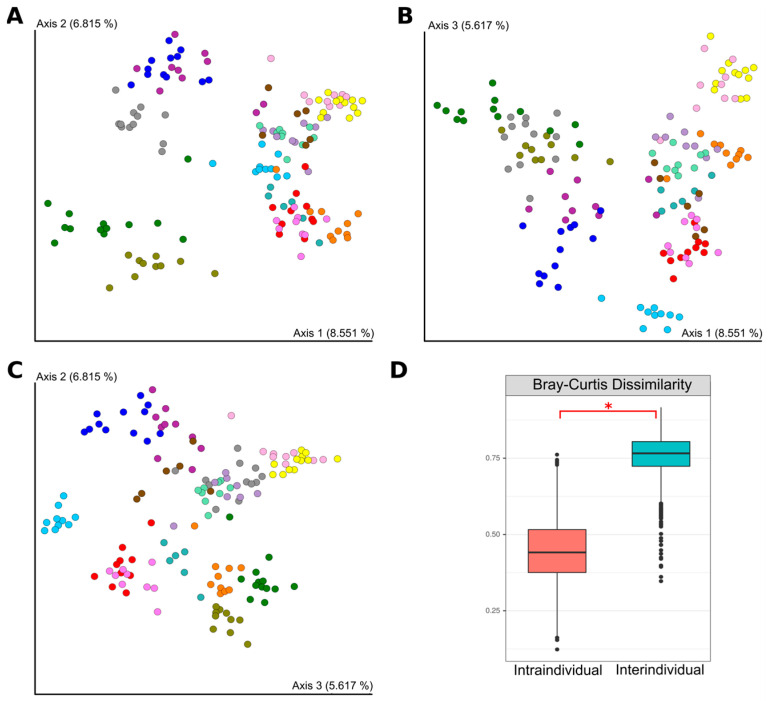
PCoA plot of the gut microbiota in healthy individuals (*n* = 15) based on Bray–Curtis dissimilarity. (**A**) Coordinate 1 and Coordinate 2. (**B**) Coordinate 1 and Coordinate 3. (**C**) Coordinate 2 and Coordinate 3. Each color represents samples from a given individual. (**D**) Intra- and inter-individual differences in Bray-Curtis dissimilarity. * *p* < 0.05 (Kruskall-Wallis test).

**Figure 5 microorganisms-12-01712-f005:**
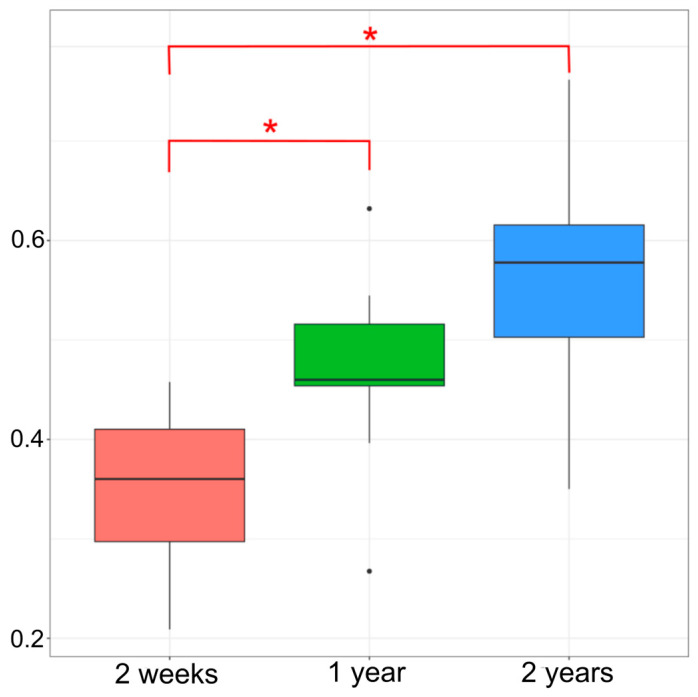
Intraindividual differences in Bray-Curtis dissimilarity for samples taken two weeks, one year, and two years apart. * *p* < 0.05 (Kruskall-Wallis test).

**Figure 6 microorganisms-12-01712-f006:**
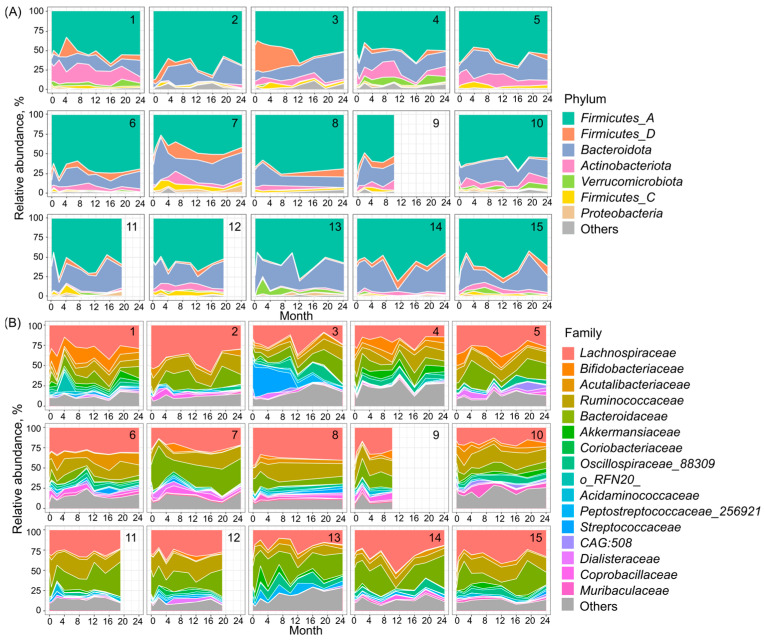
Relative abundances of (**A**) phyla and (**B**) families of bacteria in the gut microbiota of 15 healthy individuals over a 24-month period.

**Figure 7 microorganisms-12-01712-f007:**
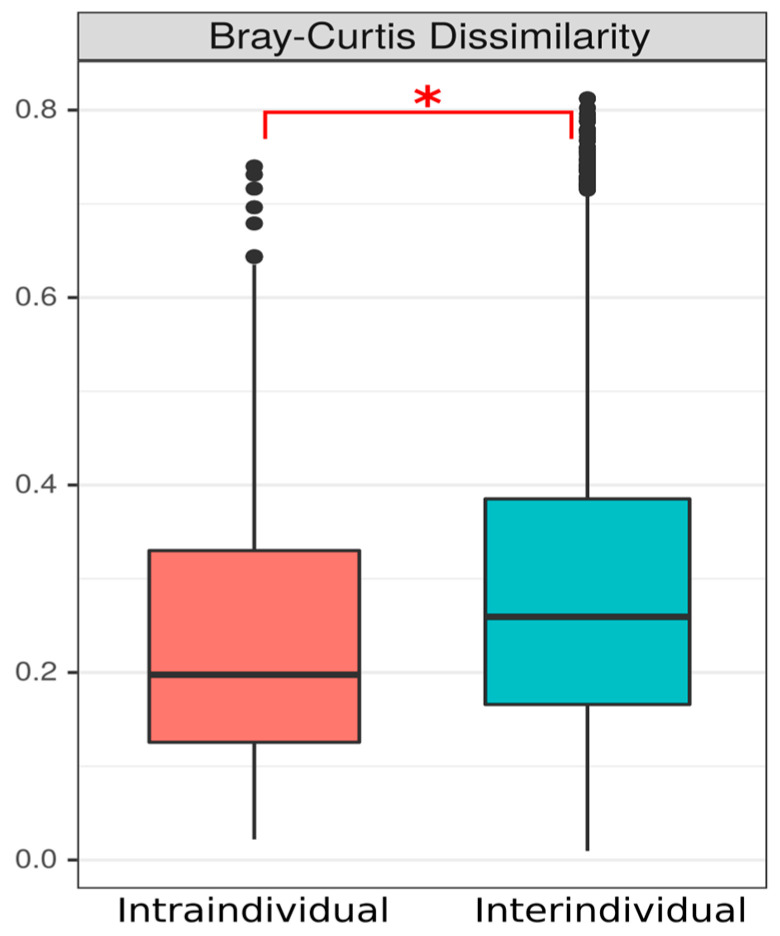
Intra-individual and inter-individual values of Bray-Curtis distances for short-chain fatty acids (acetic, propionic, isobutyric, butyric, isovaleric, valeric, isocaproic and caproic acids). * *p* < 0.05 (Kruskall-Wallis test).

**Table 1 microorganisms-12-01712-t001:** Variability of SCFAs measured in 15 healthy volunteers.

SCFAs	Min, mg/g	Max, mg/g	Mean ± SD, mg/g
C2 (acetic)	0.29	3.28	1.35 ± 0.61
C3 (propionic)	0.13	1.14	0.46 ± 0.24
iC4 (isobutyric)	0.02	0.42	0.08 ± 0.05
C4 (butyric)	0.05	1.48	0.41 ± 0.27
iC5 (isovaleric)	0.05	0.62	0.19 ± 0.09
C5 (valeric)	0.03	0.56	0.12 ± 0.07
iC6 (isocaproic)	0.0001	0.03	0.01 ± 0.01
C6 (caproic)	0.0001	0.53	0.06 ± 0.07

**Table 2 microorganisms-12-01712-t002:** Seasonal changes in bacterial taxa abundance in the gut microbiota of healthy adult volunteers.

Taxa	Relative Abundance (Mean ± SD, %)				Adjusted *p*-Values					
	Fall (F)	Spring (Sp)	Summer (S)	Winter (W)	F. vs. Sp.	F. vs. Sum.	Sp. vs. Sum.	F. vs. W.	Sp. vs. W.	Sum. vs. W.
Unclassified g. *Eggerthellaceae*	0.26 ± 0.4	0.3 ± 0.5	0.27 ± 0.5	0.27 ± 0.5	0.83	0.01 *	0.03 *	0.13	0.24	0.10
g. *Anaerobutyricum*	1.1 ± 0.6	1.7 ± 0.9	1.7 ± 0.8	1.78 ± 1	0.03 *	0.01 *	1.00	<0.001 *	0.77	0.84
g. *Pseudomonas_F*	0	0.5 ± 0.8	0.003 ± 0.01	0	<0.001 *	0.39	<0.001 *	1.00	<0.001 *	0.33
g. *Klebsiella_724518*	0.01 ± 0.03	0.2 ± 0.7	0.01 ± 0.05	0.003 ± 0.02	0.003 *	0.87	<0.001 *	0.88	<0.001 *	0.85
g. *Acinetobacter*	0	0.01 ± 0.02	0.06 ± 0.3	0	<0.001 *	0.66	<0.001 *	1.00	<0.001 *	0.68
f. *Pseudomonadaceae*	0.001 ± 0.004	0.5 ± 0.84	0.003 ± 0.01	0.0005 ± 0.003	0.003 *	0.86	0.001 *	0.70	<0.001 *	0.69
f. *Veillonellaceae*	0.03 ± 0.04	0.15 ± 0.3	0.08 ± 0.16	0.04 ± 0.1	0.11	0.28	0.38	0.77	0.04 *	0.10
f. *Moraxellaceae*	0	0.01 ± 0.02	0.06 ± 0.3	0	<0.001 *	0.66	<0.001 *	1.00	<0.001 *	0.68

Statistically significant values are shown in red (* *p* < 0.05, Kruskall-Wallis test).

## Data Availability

The data presented in this study are available on request from the corresponding author. Raw reads are deposited in the NCBI SRA under accession number PRJNA1067295 in the fastq format (https://www.ncbi.nlm.nih.gov/bioproject/1067295, accessed on 21 January 2024).

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
