# Peer review of "Two-Year Study on the Intra-Individual Dynamics of Gut Microbiota and Short-Chain Fatty Acids Profiles in Healthy Adults"

_microorganisms, 2024, doi:10.3390/microorganisms12081712_

Round 1

Reviewer 1 Report

Comments and Suggestions for Authors

General comments:

The manuscript describes the composition of the intestinal microbiota of a cohort of 15 people from the Volga region of the Russian Federation. The study suffers from some important shortcomings. Among them, the number of subjects (15) is low, and they are mainly women, not evenly distributed. The fact that all subjects were sampled following the same protocol (no controls) also reduces the relevance of the results obtained.

Nevertheless, the study also presents some interesting points. For example, the period over which the subjects were sampled is long (years), and the population comes from a region where results are not widely available, which makes the published results useful for comparison with microbial populations of the human gut microbiota from other regions.

The study complies with the ethical standards of research and has been approved by a bioethics committee, so I understand that the requirements regarding the protection of personal data and other requirements have been met.

An important aspect is that the terms “microflora”, “microbiota”, “microbiome” are used indistinctly, when in fact, in this work “microbiota” should be mentioned in all cases.

Specific comments:

Page 1, line 11. “microflora” is not a very correct expression. Please change it to “microbiota”

Page 1, line 15 “SCFAs” should be defined the first time in the abstract or citing without abbreviation.

Page 1, lines 29-31. The microbiota influences many more pathologies, such as obesity, metabolic syndrome, diabetes, allergies, or protection against viral infections. Give at least one example of each of these.

Page 2, line 64: “SCFAs” was already defined in the Introduction section.

Page 4, lines 142-143: Software and manufacturer employed for statistical analysis?

Page 4. Lines 150-160: “Of the 10 samples collected over two years, three were for the winter period, and two points were for the remaining seasons”. 3*2+3=9. It was 9 or 10 samples per subject?

Figure 6. Results from subject 9 are not complete. Why is the reason?

Page 8, Size of figures 1 and 7 are much higher than the other ones. Please correct it.

Page 8, line 245: In the all manuscript the references are separated by “,” and not “;”.

Page 9, lines 263-264: “These differences may be 263 related to population, geographic and methodological aspects.” Please, delete an space at the beginning of the phrase and add at least a reference to reinforce it.

Author Response

Dear Reviewer, 

We are grateful for your evaluation of our study, we appreciate the time that you have spent on the review.

Comments 1: Page 1, line 11. “microflora” is not a very correct expression. Please change it to “microbiota”

Response 1: Thank you for your valuable suggestion. The term "microflora" has been changed to “microbiota”, page 1, line 11.

Comments 2: Page 1, line 15 “SCFAs” should be defined the first time in the abstract or citing without abbreviation.

Response 2: We agree with this comment and removed the abbreviation page 1, line 15.

Comments 3: Page 1, lines 29-31. The microbiota influences many more pathologies, such as obesity, metabolic syndrome, diabetes, allergies, or protection against viral infections. Give at least one example of each of these.

Response 3: We accepted this comment and corrected the manuscript. We added the description of known pathologic conditions related to dysbiosis, page 1, line 31.

Comments 4: Page 2, line 64: “SCFAs” was already defined in the Introduction section.

Response 4: We accepted this comment and corrected the manuscript, page 2, line 71.

Comments 5: Page 4, lines 142-143: Software and manufacturer employed for statistical analysis?

Response 5: All statistical calculations were performed using the R v.4.2.2 programming language in the RStudio v.2023.12.1+402 program.

We added this information to the materials and methods section, page 4, line 149.

Comments 6: Page 4. Lines 150-160: “Of the 10 samples collected over two years, three were for the winter period, and two points were for the remaining seasons”. 3*2+3=9. It was 9 or 10 samples per subject?

Response 6: Thank you for noticing this inaccuracy. There were 4 points for the winter period and 2 points for the remaining seasons. We corrected the manuscript, page 4, line 167.

Comments 7: Figure 6. Results from subject 9 are not complete. Why is the reason?

Response 7: Unfortunately, some participants dropped out of the study. We decided to keep their starting points in order to expand the data on intraindividual and seasonal changes for statistics. We have a discussion related to the exclusion of patient data on page 3, line 98.

Comments 8: Page 8, Size of figures 1 and 7 are much higher than the other ones. Please correct it.

Response 8: We checked the figures. The original sizes of the figures 1 and 7 do not exceed the sizes of the other figures. If the figures do not fit the journal requirements, we hope that the editors will help us to adjust their sizes during further processing of the manuscript.

Comments 9: Page 8, line 245: In the all manuscript the references are separated by “,” and not “;”.

Response 9: We accepted this comment and corrected the manuscript.

Comments 10: Page 9, lines 263-264: “These differences may be 263 related to population, geographic and methodological aspects.” Please, delete an space at the beginning of the phrase and add at least a reference to reinforce it.

Response 10: We accepted this comment and corrected the manuscript, page 10, line 275,276.

Reviewer 2 Report

Comments and Suggestions for Authors

In the manuscript submitted to me for review entitled "Intraindividual dynamics of gut microbiota and short-chain fatty acids profiles in a two-year study of healthy adultsthe authors Anastasia Senina, Maria Markelova, Dilyara Khusnutdinova, Maria Siniagina, Olga Kupriyanova, Gulnaz Synbulatova, Airat Kayumov, Evgenia Boulygina and Tatiana Grigoryeva present a study in which they tracked the changes in the composition of the intestinal microbiota and the content of short-chain fatty acids in healthy adult volunteers over a period of 24 months.

15 volunteers were included, from whom prior consent was obtained for inclusion in the study. The research was conducted in accordance with the recommendations of the local ethics committee of Kazan Federal University, Kazan, Russia. The information presented in the manuscript contributes to a more accurate understanding of the changes in the composition of the gut microbiota during the different seasons, which would contribute after the accumulation of more research to make it possible to control the species composition of the microbiota with appropriate probiotics, prebiotics and nutritional supplements.

To support their research, the authors used 86 references that present information from studies spanning mainly the last two decades, although older studies are also presented. Nearly 1/2 of all references are from the last 5 years, indicating that the topic under consideration has also been actively researched by other authors in recent years, meaning it would also attract the attention of Microorganisms readers. I did not notice any redundant self-citations, all the references used are appropriate and necessary for the preparation of the manuscript.

My remarks and recommendations to the authors:

1. The inscriptions in some of the figures are too small and unreadable. It would be to the benefit of the readers if the authors increased the font of the inscriptions. Such figures are: Figures 4, 5 and 6.

2. Why did the authors decide to exclude pregnancy and lactation as the reason for the exclusion of research participants? These are normal conditions (not diseases) that would give interesting information - compared, both in the same participant in different seasons, after birth, transition to lactation, and in comparison with other volunteers included in the study.

3. How do the authors explain the absence of the genus Pseudomonas_F and the family Moraxellaceae in the autumn and winter samples without the use of drugs and antibiotics (according to information from the volunteers)? Not commented in the discussion.

4. There is no graphical or tabular presentation of seasonal changes in the bacterial community, only SCAFs. I think it would be helpful for readers if the authors add a figure or table to visualize the results (and in the main text of the manuscript, not in a supplementary file). In my opinion, it is precisely these results that form the basis of the manuscript.

Author Response

Dear Reviewer, 

We are grateful for your evaluation of our study, we appreciate the time that you have spent on the review.

Comments 1: The inscriptions in some of the figures are too small and unreadable. It would be to the benefit of the readers if the authors increased the font of the inscriptions. Such figures are: Figures 4, 5 and 6.

Response 1: We agree with this comment. All figures and their inscriptions have been enlarged and standardized.

Comments 2: Why did the authors decide to exclude pregnancy and lactation as the reason for the exclusion of research participants? These are normal conditions (not diseases) that would give interesting information - compared, both in the same participant in different seasons, after birth, transition to lactation, and in comparison with other volunteers included in the study.

Response 2: Pregnancy and lactation affect the composition of the gut microbiota (DOI: 10.1097/NMC.0000000000000372). Our sample collection is small, so we excluded additional factors to assess the dynamics of microbiota composition. It will be interesting to include such participants in a further study.

Comments 3: How do the authors explain the absence of the genus Pseudomonas_F and the family Moraxellaceae in the autumn and winter samples without the use of drugs and antibiotics (according to information from the volunteers)? Not commented in the discussion.

Response 3: Unfortunately, there is a lack of studies on the bacterial genera that are associated with the human lifestyle. We suppose that the seasonal variation and availability of some products may influence the microbiota, resulting in  changes in autumn and spring samples. In addition, the changes were related to  minor representatives (less than 1.5% of the community in average). So the absence of some genera  and families may be associated with a decrease in their proportion in the community below the detection threshold.

Comments 4: There is no graphical or tabular presentation of seasonal changes in the bacterial community, only SCAFs. I think it would be helpful for readers if the authors add a figure or table to visualize the results (and in the main text of the manuscript, not in a supplementary file). In my opinion, it is precisely these results that form the basis of the manuscript.

Response 4: We have added a table 3 on page 8, line 239.

Round 2

Reviewer 1 Report

Comments and Suggestions for Authors

All my comments and sugestions were correctly addressed. The justify and tabulations of Table 3 and referentes were not correctly addresed and must be correctly before it's final aceptation

Author Response

Comment 1: All my comments and sugestions were correctly addressed. The justify and tabulations of Table 3 and referentes were not correctly addresed and must be correctly before it's final aceptation

Response 1: Dear Reviewer,
Thank you for your comment. We have changed the reference to the Table 3 in the text. The table has also been improved. We have corrected the references, page 12, line 378.